# Comparison of the Prevalence and Location of Trigger Points in Dressage and Show-Jumping Horses

**DOI:** 10.3390/ani15111558

**Published:** 2025-05-27

**Authors:** Karine Portier, Camilla Schiesari, Lisa Gauthier, Lin Tchia Yeng, Denise Tabacchi Fantoni, Maira Rezende Formenton

**Affiliations:** 1VetAgro Sup (Campus Vétérinaire), Centre de Recherche et de Formation en Algologie Comparée (CREFAC), University of Lyon, 69280 Marcy l’Etoile, France; karine.portier@vetagro-sup.fr (K.P.); lisagauthier97@gmail.com (L.G.); 2Centre National de la Recherche Scientifique (CNRS), Institut National de la Santé et de la Recherche Médicale (INSERM), Centre de Recherche en Neurosciences de Lyon (CRNL), University of Lyon, U1028 UMR 5292, Trajectoires, 69500 Bron, France; 3School of Medicine, Institute of Orthopedics and Traumatology, University of São Paulo, São Paulo 05403-010, Brazil; lintyeng@uol.com.br; 4School of Veterinary Medicine and Animal Science, University of São Paulo, São Paulo 05508-270, Brazil; dfantoni@usp.br (D.T.F.); mairaformenton@gmail.com (M.R.F.)

**Keywords:** myofascial trigger point, equine pain, thermography, analgesia, muscle pain

## Abstract

Myofascial trigger points are muscular small, indurated, painful areas that are known to cause discomfort and reduced performance in humans, but their role in horses is less well understood. In this study, we examined fourteen sport horses (seven dressage, seven show-jumping) to identify and localize trigger points by manual palpation. The body temperature at the trigger points was measured. Pain induced by palpation was scored using the horse facial grimace scale. Dressage horses had more trigger points on their necks, while show-jumpers had more on their backs and rumps. The trigger points were hotter and more painful compared to a normal area of the muscle. These findings suggest that the type of training influences the distribution of muscle tension points. Identifying and treating these areas could help improve horse welfare and athletic performance.

## 1. Introduction

Myofascial pain is a musculoskeletal disorder characterized by pain in the muscle and its surrounding connective tissue, known as fascia [1]. This condition usually affects individuals between the ages of 27 and 50 and up to 85% of the general population [2]. Myofascial pain syndrome (MPS) has also been described in horses [3]. It can be clinically expressed by lameness, back pain, and reduced sporting performance, leading to economic losses [4].

The MPS is characterized by trigger points, usually called Myofascial Trigger Points (MTrPs). They are found within a band of muscle tension and are characterized by a nodule hypersensitive to palpation in humans [5], dogs [6,7], and horses [4]. Signs associated with MTrPs may include local and/or referred pain as well as paresthesia leading to reduced range of motion. Patients often describe the pain as diffuse and deep, accompanied by a burning sensation and areas of radiation [8].

Studies of MPS in horses are rare and involve small samples. However, a study by Macgregor and Von Schweinitz [9] investigated the characteristics of MTrPs in horses, identifying local twitch responses and electrophysiological activity, using needle electromyography. These characteristics were similar to those of MTrPs observed in humans [9,10].

In humans, the risk factors contributing to MPS include ergonomic factors (such as excessive activities and abnormal posture), structural factors (such as spondylosis, scoliosis, and osteoarthritis), and systemic factors (such as vitamin D deficiency, hypothyroidism, and iron deficiency) along with traumatic events [11]. In sports horses, the effort required, joint stress from repetitive movements, the use of tack, and the rider’s influence can also lead to muscular tension [4,12].

Several techniques have been described, in combination with special palpation techniques, to locate MTrPs. Thermography has been used in humans and horses as a complementary diagnostic tool for MPS [13,14]. Pressure algometry has been described as a method of quantifying localized pain at the level of the MTrPs [15]. Additionally, signs of pain in horses can be assessed by observation of various elements of facial expression [16]. However, to our knowledge, none of these techniques have been fully validated in horses.

In human athletes, the prevalence of myofascial pain syndrome is associated with the type of sport practiced [17,18]. In equestrian sports, different disciplines require specific training, targeting various muscle groups [19].

We therefore hypothesized that the prevalence and location of muscle tension points (MTrPs) differ between dressage and show-jumping horses.

This study aimed to compare the prevalence and location of myofascial pain in the back, neck and head of show-jumping and dressage horses. The second objective was to evaluate the effectiveness of thermography and pressure algometry in characterizing MTrPs in horses. In addition, facial expressions of pain during palpation of MTrPs were examined.

## 2. Materials and Methods

This single-center experimental study was carried out in a private (Ecurie de Grange Neuve in Marcy l’Etoile, France) close to the Vetagro Sup equine clinic (Lyon Veterinary School, Lyon, France). The horses were housed in straw-bedded boxes and went out to the paddock every day for 4 h. They were fed pellets three times a day and unlimited hay. They were ridden for 1 h every day in training and regularly participated in official competitions in France (on average, one competition per month between February and October). The study was approved by VetAgro Sup’s Ethical Committee (Number 2143, 24 June 2021). Written consent was obtained from each owner before their horse was included in the study.

### 2.1. Animals

The horses included in the study were in good health and trained exclusively for show-jumping or dressage disciplines. They were allocated to groups based on their respective discipline. Inclusion criteria were as follows: horses that had a normal clinical and orthopedic examination, aged between 4 and 15 years, and trained for show jumping or dressage while actively competing at the time of the study, and whose owners gave their agreement.

Exclusion criteria were as follows: horses that showed lameness, had competed on the same day or the day before, were uncooperative or stressed, had undergone surgery within the same year, were receiving analgesic treatment, or had dermatological conditions.

The horses underwent a clinical and orthopedic examination to confirm their good health and ensure that they were free from lameness. The clinical examination consisted of an inspection of the body, cardiac and respiratory auscultation, and measurement of rectal temperature. The orthopedic examination consisted of observation of the horse in movement at the walk and trot in a straight line and on the circle, a pinch test of the feet, a girth test at the trot, and flexion tests of the joints of the four limbs at the trot. Grid lines (10 cm × 10 cm) were drawn with chalk on both sides of the head, neck, and back, each cell corresponding to a muscle (or part of a muscle).

### 2.2. Outcome Measures

The primary outcome of this study was to determine the prevalence and distribution of myofascial trigger points (MTrPs) in different muscle groups of show-jumping and dressage horses. Secondary outcomes included differences in pain response, assessed using the Horse Grimace Scale (HGS), and variations in thermographic and algometric measurements at MTrP locations compared to control points.

### 2.3. Trigger Point Identification

Two evaluators palpated the horses on both sides of the head, neck, and back. Each assessor was unaware of the other’s evaluations, and both were blinded to the type of competition the horse was participating in (show jumping or dressage). One of the examiners was a veterinary surgeon specialist in animal physiotherapy. The other examiner is a veterinary specialist from the European College of Veterinary Anaesthesia and Analgesia who had performed equine orthopedics in equine practice for 10 years. Palpation was carried out in the same sequence by both assessors: head, neck, and back on the previously drawn grid. The assessors palpated each cell and then recorded the cell number to identify the location of the trigger point.

Palpation began with the masseter muscle. The neck was palpated, focusing on the brachiocephalicus muscle (dorsal and ventral), the splenius, serratus ventralis, and trapezius pars cervicalis muscles. The back was palpated along the trapezii thoracis, longissimi thoracis, longissimus lumborum, and serratus dorsalis muscles. The final zone, the croup, was palpated along the gluteus medius muscle.

The criteria for identifying a MTrP were based on the literature. They primarily involved the palpation of hypercontractile nodules, with a typical “twitch” response, which is defined as a transient visible or palpable contraction of the muscle fibers in response to pressure [9,20]. The horse could also vocalize, look at the painful area or the assessor, arch its body, tap, or avoid contact [21].

Only the points identified as MTrPs by the two evaluators were subsequently studied.

### 2.4. Pressure Algometry

For each identified MTrP, the pressure algometer (Instrutherm^®^, Dd-500, São Paulo, Brazil) was applied perpendicularly to the skin and pressure was slowly increased at a rate of approximately 1 kg/cm^2^/s until the animal showed signs of discomfort/pain (including a contractile response, an attempt to escape pressure, and vocalizations) [22]. The pressure was recorded in kg/cm^2^. Two evaluators conducted the measurements twice and the correlation between their results was subsequently calculated.

### 2.5. Thermographic Evaluation

After the algometric measurements, horses were kept in a room for 20 min to acclimate to the ambient temperature. The thermograph (FLIR E40^®^, Wilsonville, OR, USA) was calibrated to the room temperature. Thermographic images were captured from two meters on both sides of the head, neck, and back (lateral and dorsal views). The images were taken freehand, a single image per point, when the values had stabilized. Temperature measurements were conducted in the squares previously marked as containing MTrPs. The temperature control was conducted in an area where the two assessors had not identified any MTrPs. The data were processed with a specific software (FLIR Tools^®^, version 2.1 Wilsonville, OR, USA). A rainbow palette was used, with red representing hot temperatures and blue/black indicating cold temperatures. The temperature range was set at 15–40 °C and emissivity at 0.98. Once the area had been located, the average, minimum, and maximum temperatures of the quadrilaterals (16 × 16 pixels) were measured, with an average of 256 pixels. The rectal temperature of each horse was measured with a thermometer for control purposes.

### 2.6. Horse Grimace Scale

During MTrPs palpation, a video was taken of the horse’s face. After analysis, screenshots were taken before palpation of the trigger point (with the hand resting on the horse) and during the palpation (applying pressure). The photos were anonymized and sent to 3 observers, who scored them using a previously published horse grimace pain scale written in French [23]. Observer 1 was neither a veterinarian nor a rider but had been trained in the use of HGS. Observer 2 was a veterinarian but not a rider. Observer 3 was a veterinarian specializing in algology and a horse rider. Observers 2 and 3 had not been trained to use the HGS.

No adverse event was reported during the study.

### 2.7. Statistical Analysis

A power calculation was performed to calculate the minimum sample size required to have an 80% chance of identifying, with a risk of 5%, an average difference of 5 MTrPs between dressage and show-jumping horses. This calculation indicated that seven horses were required in each group.

Data distributions were tested for normality using the Shapiro–Wilk test. Data are expressed as mean ± standard deviation for normally distributed data or median [min, max] for non-normally distributed data as appropriate. Differences were considered significant if *p* < 0.05.

The distribution of the MTrPs by body area was calculated by dividing the total number of MTrPs identified in each specific area by the overall number of MTrPs per horse.

The prevalence of MTrPs in a specific muscle (or group of muscles) was determined by counting the number of horses that had at least one MTrP in that muscle (or group of muscles) and dividing this by 7, the total number of horses in each group (with one muscle/group of muscles assessed per horse). This was performed for each side of the 7 horses in each group. For example, if MTrPs were identified in the left gluteus muscle of 4 dressage horses, the prevalence of MTrPs in the left gluteus muscle of dressage horses would be calculated as 4/7, resulting in 57%. The calculation was based on data collected by each evaluator and their intersection, with each muscle assessment treated as an independent measure. A descriptive comparison of the prevalence was carried out.

Trigger point temperatures obtained through thermography were compared to the control temperature points using the Wilcoxon rank-sum test.

Concerning the algometer, the degree of correlation between the data obtained by the two evaluators was assessed using a Spearman correlation test.

The facial grimace scores obtained during the palpation of the MTrPs were compared to the control scores using the Wilcoxon signed-rank tests.

## 3. Results

### 3.1. Horses

Seven show-jumping horses (9 ± 1 years, three females, four geldings) and seven dressage horses (10 ± 3 years, four females, two geldings, one stallion) entered the study.

### 3.2. MTrPs Distribution and Prevalence

At least one MTrP was identified on each horse regardless of its sporting activity.

Both groups had a high prevalence (>60%) of MTrPs in the back. Dressage horses had a higher prevalence of MTrPs in the neck area (17%) than show-jumping horses (3%). Conversely, show-jumping horses had a higher prevalence of MTrPs in the rump area (30%) than in dressage horses (17%) (Table 1).

Both groups had the highest prevalence of MTrPs in the *longissimus thoracis* muscle (71% in dressage horses and 86% in show-jumping horses). In dressage horses, the brachiocephalicus muscle showed the highest prevalence of MTrPs (dorsal part 29%, ventral part 14%), especially in the cervical region, while in show-jumping horses, MTrPs were most prevalent in the *gluteal* and *trapezius pars thoracica* muscles (86% and 71%, respectively) (Figure 1).

### 3.3. Algometry

Muscle tension values obtained by algometry were not correlated between the two observers (r = 0.20, *p* = 0.32).

### 3.4. Thermography

The MTrPs had significantly higher minimum (34.2 [29.5–37.6] °C vs. 33.0 [31.8–36.0] °C), average (35.0 [30.4–38.2] °C vs. 33.8 [32.6–36.3] °C) and maximum (35.6 [31.0–38.8] °C vs. 34.8 [33.6–36.6] °C) temperatures than the control points (*p*-value = 0.009, 0.012, 0.003, respectively) (Figure 2).

### 3.5. Horse Grimace Score

The three observers gave a significantly higher overall horse grimace score when the MTrPs were palpated than when the hand was just placed on it (control) (16 [0–24] vs. 6 [0–19], *p* = 0.004).

Observer 1, 2, and 3 scores during MTrPs vs. Control were

Observer 1: 3 [0–7] vs. 0 [0–3], *p* = 0.005;Observer 2: 6 [0–10] vs. 3 [0–8], *p* = 0.04;Observer 3: 8 [0–11] vs. 2 [0–8], *p* = 0.008.

Observer 1 gave lower scores for facial expressions photographed during control and during the MTrPs palpation than observers 2 and 3 (*p* = 0.008 and 0.041, respectively), (*p* = 0.012 and 0.003, respectively) (Figure 3).

The first line of the boxplot indicates the lower interquartile value, the middle line indicates the median, and the third line indicates the upper interquartile value. The ‘whiskers’ indicate the maximum and minimum, excluding the outliers represented by the empty circles.

The three observers’ grimace scale scores were significantly higher in show-jumping horses during the palpation of the MTrPs than during the control (17 [15–23] vs. 6 [01–19], p = 0.015), whereas they were not significantly different between palpation and control in dressage horses (14 [00–24] vs. 4 [00–12], *p* = 0.100) (Figure 4).

## 4. Discussion

The findings of this study indicate that the distribution of myofascial trigger points (MTrPs) significantly differs between show-jumping horses and dressage horses.

In humans, physical activity also influences the location of MTrPs. The *trapezius pars cervicalis* and *trapezius pars thoracica* had the highest prevalence of MTrPs in fighting sports athletes. However, the distribution of MTrPs in muscles varies according to the sports practiced. In judokas, for example, MTrPs are most often found in the trapezius, *quadratus lumborum*, and *quadriceps femoris* muscles, while in boxers they are mainly found in the *trapezius and brachioradialis* muscles [18]. The cervical region and shoulder muscles are also particularly affected by overuse and/or prolonged uncomfortable postures [24]. For example, myofascial pain associated with trigger points in the upper trapezius is among the most frequent musculoskeletal pain syndromes in the shoulder area of food service workers who perform repetitive movements of the upper limbs [25]. In horses, MTrPS have been identified in various muscles; however, to our knowledge, the influence of the type of training on the location of MTrPs has not yet been reported. Indeed, MTrPs have been detected in the *cervical and pectoral* muscles, in the *brachiocephalic* muscle [9], in the axillary region of the cranial portion of the *ascending pectoralis* muscle [4], the *triceps*, the *iliotibial* muscle, the *paraspinal* muscles of the *neck*, and the *gluteal* and *hip* muscles as well as in the *trapezius* [3].

The diagnosis of the myofascial pain syndrome is mainly clinical, involving MTrPs palpation [26]. However, the subjectivity of clinical assessment implies the need for other, more objective methods. In humans, thermography and pressure algometry are mentioned as objective methods of MTrPs detection [27,28]. These tools have also been used in horses to identify and localize pain [13,20,29,30]. Nevertheless, to our knowledge, no study has used these techniques to identify MTrPs in horses.

Skin temperature at MTrPs was higher than in MTrPs-free tissue. On the contrary, chronic pain sites can have a reduced temperature due to the sympathetic blocking mechanism in humans [31]. These results must be interpreted with caution, as thermography is still mainly used experimentally and has long been considered a marginal method. However, it is now increasingly recognised as a diagnostic tool for measuring body tissue temperature in various pathological situations, such as inflammation, infection, and tumors [32,33], as well as for monitoring thermal homeostasis in horses during transport [34]. It has even been shown that thermography can be useful for monitoring the development of MTrPs in humans, as the microcirculation around these points is altered and subject to vegetative changes [35]. However, this is disputed by studies that show no correlation between skin temperature and the presence of MTrPs [36]. These contradictions may be explained by the fact that the technique must be used according to certain well-defined procedures, such as control of ambient temperature, patient preparation and acclimatization, camera orientation and distance, minimum thermal resolution, accuracy, and thermal sensitivity. The training or certification of the personnel assigned to perform thermography is also important [32].

The benefits of pressure algometry are that it is a relatively inexpensive, portable, repeatable, and semi-objective technique to assess nociceptive thresholds in horses [29]. A literature review describes pressure algometry as an objective means of quantifying musculoskeletal pain in horses [37]. However, in our study, the inter-observer correlation for pressure algometry measurements was poor. Variations in handling the device and applying consistent pressure likely contributed to this. In general, intra-examiner repeatability is reported to be good and inter-examiner repeatability often depends on the level of operator experience with using the instrument [38]. Device handling is an important feature to consider when using hand-held devices. The size, shape, and ability to grasp the device and generate the consistent forces needed to produce reliable measurements is critical. As nociceptive thresholds and the response to noxious stimuli are individually based, pressure algometry may be more useful for the quantification of intra-subject reactions, rather than a method for assessing nociceptive differences between horses [29]. Our results highlight the need for better training and standardized guidelines for the use of algometry in horses. Guidelines for proper application are also needed to ensure accurate and repeatable results. Algometry should be recorded with minimal restraint in a quiet area in a routine and systematic manner to limit any external factors from influencing the horse’s attention or mental state [39].

Palpation of these points caused pain that could be recognized by the horses’ facial expressions, regardless of the observer’s experience. However, dressage horses seemed to express less pain than show-jumping horses.

The horse’s facial expression is one of the forms of pain identification and assessment reported in the literature [16,23]. In the present study, the observers marked photos and not videos, which was shown to be less effective to score the facial expression of pain [40]. It has also been demonstrated that training and previous experience using the facial rating scale may confer differences in assessment [40]. Despite all these observations, all three observers, regardless of their professional experience, recognized the facial expressions associated with MTrPs palpation compared to controls.

Dressage horses did not seem to express their pain as much, perhaps their training makes them more stoic. Stress/fear and animal characteristics of the different sports may confer this difference, as also mentioned by Dalla Costa et al. (2017) [41,42]. This raises questions about how behavioral conditioning affects pain expression and detection, which could vary significantly across equestrian disciplines.

The study involved only 14 horses (7 in each group: dressage and show-jumping). A sample this small is unlikely to capture the full variability of MTrP prevalence and distribution in the general population of sport horses.

Detection of MTrPs relied on manual palpation, a method that is inherently subjective. Evaluators may vary in sensitivity and interpretation, which introduces bias. While two evaluators performed palpation, and results were cross-checked, this does not eliminate variability. Results might be less reproducible in different settings or with other evaluators.

While thermography and pressure algometry are used in humans, their validation for detecting MTrPs in horses is limited or non-existent. Without validation studies, the reliability and clinical relevance of these tools in equine myofascial pain diagnosis remains uncertain. Despite the human literature, which suggests a minimum of 15 min after palpation to collect thermography images [43], it is possible that the 20 min waiting period for acclimatization to room temperature, after the algometry tests and manual palpation of the MTrPs, was not sufficient to soothe any residual inflammation.

The study only focused on dressage and show-jumping horses. It did not include other equestrian disciplines or untrained horses. As a consequence, the findings might not apply to horses in different disciplines or those with different levels of training and physical demands.

This study captured data at a single point in time, without exploring how MTrPs develop, resolve, or respond to treatment over time. The study provides a static snapshot but does not explore the dynamic nature of myofascial pain, which limits the ability to draw conclusions about causality or the progression of MTrPs.

## 5. Conclusions

This study demonstrated that the location of myofascial trigger points (MTrPs) significantly differs between jumping horses and dressage horses, likely reflecting the specific muscular demands of each discipline. MTrPs were more frequent in the neck muscles of dressage horses, while the back and croup muscles were more affected in show-jumping horses. These findings confirm the influence of the type of sporting activity on the distribution of MTrPs, as previously observed in human athletes. The findings of this study open perspective for better recognition of the muscular specificities of different equestrian disciplines and emphasize the need for more reliable diagnostic tools to assess and treat myofascial pain in horses, thereby improving their welfare and sporting performance. Further studies with larger sample sizes, including horses from different equestrian disciplines, and longitudinal designs are needed to validate these findings and improve the reliability of MTrPs detection methods in horses.

## Figures and Tables

**Figure 1 animals-15-01558-f001:**
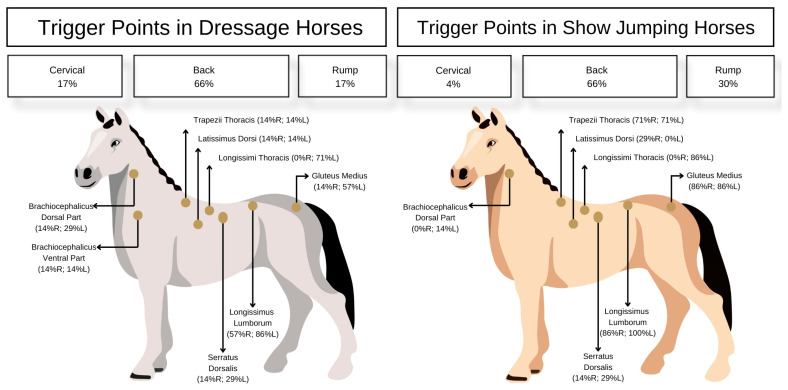
Prevalence of MTrPs in 7 dressage and show-jumping horses.

**Figure 2 animals-15-01558-f002:**
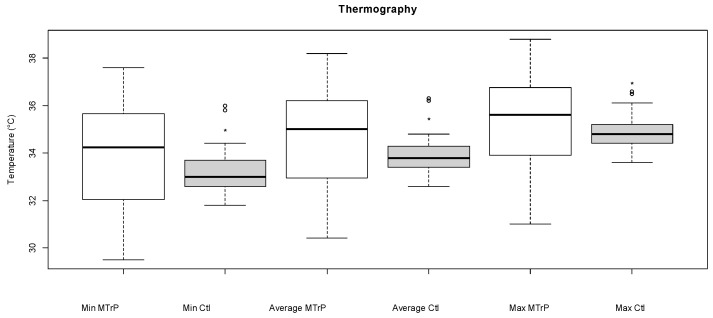
Boxplots representing minimum (Min), maximum (Max), and average (Average) temperatures (°C) were measured by thermography of trigger (MTrP, white) or control points (Ctl, grey) in 14 horses. The first line of the boxplot indicates the lower interquartile value, the middle line indicates the median, and the third line indicates the upper interquartile value. The ‘whiskers’ indicate the maximum and minimum, excluding the outliers represented by the empty circles.* Min, average, or max temperature significantly different from the min, average, or max temperature of the MTrP (*p* < 0.05).

**Figure 3 animals-15-01558-f003:**
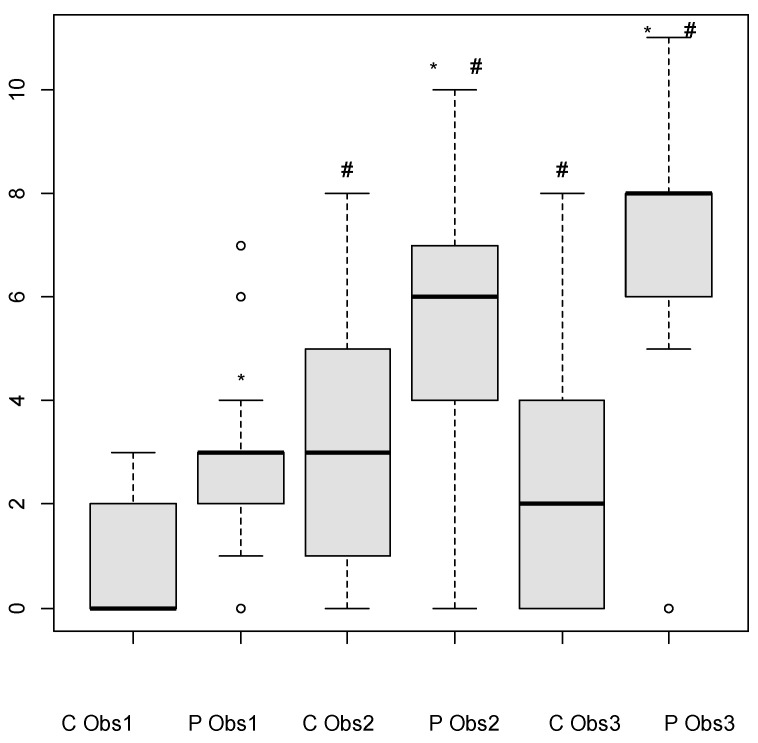
Pain scores given by 3 observers using the Horse Grimace Scale (French version) looking at the facial expression of 14 horses during the palpation of a trigger point (P) or hand just placed on the body (C). Observer 1 (Obs1) was neither a veterinarian nor a rider but had been trained in the use of HGS. Observer 2 (Obs 2) was a veterinarian but not a rider. Observer 3 (Obs 3) was a veterinarian specializing in algology and a horse rider. Observers 2 and 3 had not been trained to use the HGS. * Significantly different from C for the same observer. # Significantly different from observer 1.

**Figure 4 animals-15-01558-f004:**
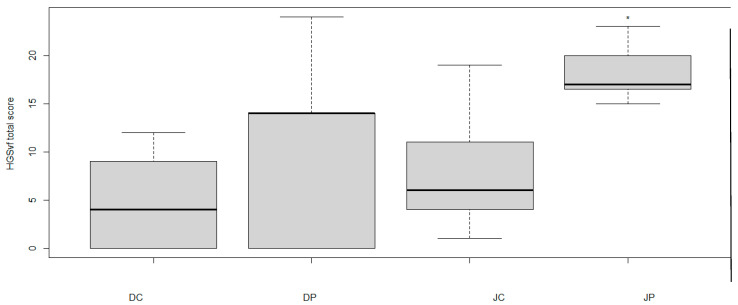
Pain scores given by 3 observers using the Horse Grimace Scale (French version) looking at the facial expressions of 7 dressage horses (D) and 7 show-jumping horses (J) during the palpation of a trigger point (P) or hand-just placed on the body (C). * Significantly different from C. The first line of the boxplot indicates the lower interquartile value, the middle line indicates the median, and the third line indicates the upper interquartile value. The ‘whiskers’ indicate the maximum and minimum.

**Table 1 animals-15-01558-t001:** Distribution of MTrPs in the body parts of 7 dressage and show-jumping horses.

	Dressage		Show Jumping	
Area	Number of MTrP ^1^	Distribution by Area	Number of MTrP	Distribution by Area
Neck	8	17%	3	4%
Back	31	66%	48	66%
Rump	8	17%	22	30%
Total	47	100%	73	100%

^1^ Legend: Myofascial trigger points.

## Data Availability

Data are contained within the article.

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
