# Peer review of "Comparison of the Prevalence and Location of Trigger Points in Dressage and Show-Jumping Horses"

_animals, 2025, doi:10.3390/ani15111558_

Round 1

Reviewer 1 Report

Comments and Suggestions for Authors

The manuscript entitled “Comparison of the prevalence and location of trigger points in dressage and show-jumping horses” investigate the Myofascial pain syndrome in show jumping and dressage horses.

The present study is very interesting and innovative. it is structured quite well and written in an understandable and easy-to-follow manner. However in my opinion, there are several gaps especially in the section of materials and methods which are an important element for the whole management of the experiment, if these are filled and deepened the work could be accepted for publication. In attachment specific comment could be found.

Author Response

Please see the attachment with our point-by-point responses to your comments.
All modifications in the manuscript have been highlighted for clarity.

Thank you again for your constructive feedback.

Best regards,
Dr. Camilla Schiesari (on behalf of all authors)

Reviewer 2 Report

Comments and Suggestions for Authors

I have been fascinated to read this manuscript and would like to commend the authors on their study. I have only a few minor comments and thoughts.

Materials and Methods

Section 2.2 Outcome measures

Lines 108-111: Secondary measures should also include pressure algometry.

Section 2.3 Trigger Point Identification

Line 122: ‘longissimus thoracicis’. Should this be ‘longissimus thoracis’?

Section 2.4 Pressure algometry

Lines 135-137: ‘Compression pressure was increased by approximately 1 to 2mm per second’; this is the rate of advancement of the algometer ‘pin’, and not the rate of pressure application. Could the authors give an approximate rate of pressure application, e.g. in Newtons per square cm per second?

Section 2.5 Thermographic evaluation

Line 143: Was 20 minutes after algometry (which followed manual palpation and identification of trigger points), long enough for any residual ‘irritation’ (mild local vasodilation/inflammation) following manual palpation of the MTrPs, to have subsided?

Section 2.7 Statistical Analysis

Line 183: The authors use the abbreviation ‘TP’ (trigger point) here, but elsewhere they use MTrP. I think, for consistency, ‘MTrP’ could be used here also.

Results

Section 3.2 MTrPs Distribution and Prevalence

Lines 213-214: The statement: ‘Muscle tension values obtained by algometry… p=0.32).’, should be deleted from here, as it is repeated under “Section 3.3 Algometry”.

Figure 1:

‘Longissimus thoracicae’ and ‘Trapezius thoracicae’ – should be ‘Longissimi thoracis’ and ‘Trapezii thoracis’.

‘Brachiocefalicus’ should be, ‘Brachiocephalicus’.

I am fascinated by the left-right differences in number of trigger points identified. It seems that a greater number of MTrPs were identified on the left sides of the horses - were both evaluators right-handed?, or could it be something to do with riders always mounting from the left side? and/or were horses always trained/worked more on one rein than the other?

Discussion

Line 275: ‘laryngeal portion of the ascending pectoralis muscle’. I was not aware that the deep pectoral muscle had a laryngeal portion? Please could the authors check this?

Line 279: ‘myofascial syndrome’: I think the authors should state ‘myofascial pain syndrome’ to be more consistent with the rest of the manuscript.

Lines 301-317: There is another paper which describes the potential pitfalls when performing pressure algometry – this possibly should be included in the references?
Pongratz U and Licka T (2017) Algometry to measures pain threshold in the horse’s back - an in vivo and in vitro study. BMC Veterinary Research (2017) 13: 80. Doi: 10.1186/s12917-017-1002-y

Line 324: ‘In the present study The observers..’, should be, ‘In the present study the observers’ (that is, there is no need for a capital ‘T’ in the second ‘the’).

Author Response

Please see the attachment with our point-by-point responses to your comments.
All modifications in the manuscript have been highlighted for clarity.

Thank you again for your constructive feedback.

Best regards,
Camilla Schiesari (on behalf of all authors)

Round 2

Reviewer 1 Report

Comments and Suggestions for Authors

The manuscript has now been succesfully improved and is ready for pubblication

Author Response

Dear Reviewer,

Thank you very much for your positive feedback and for taking the time to review our manuscript. We are pleased to see that the revised version has met your expectations and is now considered ready for publication.

Kind regards, 

Dr. Schiesari and all the authors